# INPL: PSEUDO-LABELING THE INLIERS FIRST FOR IMBALANCED SEMI-SUPERVISED LEARNING

**Zhuoran Yu**   **Yin Li**   **Yong Jae Lee**
University of Wisconsin-Madison
{zhuoran.yu, yin.li}@wisc.edu
yongjaelee@cs.wisc.edu

## ABSTRACT

Recent state-of-the-art methods in imbalanced semi-supervised learning (SSL) rely on confidence-based pseudo-labeling with consistency regularization. To obtain high-quality pseudo-labels, a high confidence threshold is typically adopted. However, it has been shown that softmax-based confidence scores in deep networks can be arbitrarily high for samples far from the training data, and thus, the pseudo-labels for even high-confidence unlabeled samples may still be unreliable. In this work, we present a new perspective of pseudo-labeling for imbalanced SSL. Without relying on model confidence, we propose to measure whether an unlabeled sample is likely to be "in-distribution"; i.e., close to the current training data. To decide whether an unlabeled sample is "in-distribution" or "out-of-distribution", we adopt the energy score from out-of-distribution detection literature. As training progresses and more unlabeled samples become in-distribution and contribute to training, the combined labeled and pseudo-labeled data can better approximate the true class distribution to improve the model. Experiments demonstrate that our energy-based pseudo-labeling method, **InPL**, albeit conceptually simple, significantly outperforms confidence-based methods on imbalanced SSL benchmarks. For example, it produces around 3% absolute accuracy improvement on CIFAR10-LT. When combined with state-of-the-art long-tailed SSL methods, further improvements are attained. In particular, in one of the most challenging scenarios, InPL achieves a 6.9% accuracy improvement over the best competitor.

## 1 INTRODUCTION

In recent years, the frontier of semi-supervised learning (SSL) has seen significant advances through pseudo-labeling (Rosenberg et al., 2005; Lee et al., 2013) combined with consistency regularization (Laine & Aila, 2017; Tarvainen & Valpola, 2017; Berthelot et al., 2020; Sohn et al., 2020; Xie et al., 2020a). Pseudo-labeling, a type of self-training (Scudder, 1965; McLachlan, 1975) technique, converts model predictions on unlabeled samples into soft or hard labels as optimization targets, while consistency regularization (Laine & Aila, 2017; Tarvainen & Valpola, 2017; Berthelot et al., 2019; 2020; Sohn et al., 2020; Xie et al., 2020a) trains a model to produce the same outputs for two different views (e.g, strong and weak augmentations) of an unlabeled sample. However, most methods are designed for the *balanced* SSL setting where each class has a similar number of training samples, whereas most real-world data are naturally *imbalanced*, often following a long-tailed distribution. To better facilitate real-world scenarios, imbalanced SSL has recently received increasing attention.

State-of-the-art imbalanced SSL methods (Kim et al., 2020; Wei et al., 2021; Lee et al., 2021) are build upon the pseudo-labeling and consistency regularization frameworks (Sohn et al., 2020; Xie et al., 2020a) by augmenting them with additional modules that tackle specific imbalanced issues (e.g., using per-class balanced sampling (Lee et al., 2021; Wei et al., 2021)). Critically, these methods still rely on *confidence*-based thresholding (Lee et al., 2013; Sohn et al., 2020; Xie et al., 2020a; Zhang et al., 2021) for pseudo-labeling, in which only the unlabeled samples whose predicted class confidence surpasses a very high threshold (e.g., 0.95) are pseudo-labeled for training.

Confidence-based pseudo-labeling, despite its success in balanced SSL, faces two major drawbacks in the imbalanced, long-tailed setting. First, applying a high confidence threshold yields significantly

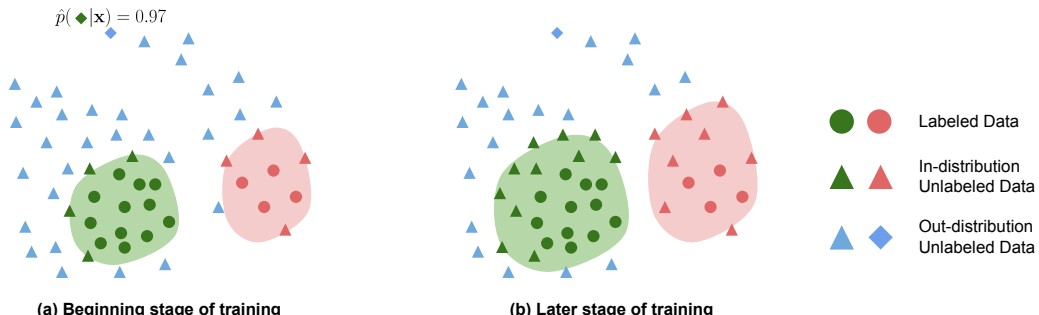

Figure 1: We illustrate the idea of InPL with a toy example with one head class (green) and one tail class (red). (a) At the beginning of training, only a few unlabeled samples are close enough to the training distribution formed by the initial labeled data. Note that with a confidence-based approach, the diamond unlabeled sample would be added as a pseudo-label for the green class since the model's confidence for it is very high (0.97). Our InPL instead ignores it since its energy score is too high and is thus considered out-of-distribution at this stage. (b) As training progresses, the training distribution is evolved by both the initial labeled data and the pseudo-labeled "in-distribution" unlabeled data, and more unlabeled data can be included in training. In this example, with our approach InPL, the diamond sample would eventually be pseudo-labeled as the red class.

lower recall of pseudo-labels for minority classes (Wei et al., 2021), resulting in an exacerbation of class imbalance. Lowering the threshold can improve the recall for tail classes but at the cost of reduced precision for other classes (see analysis in Section 4.4). Second, prior studies (Szegedy et al., 2014; Nguyen et al., 2015; Hein et al., 2019) show that softmax-based confidence scores in deep networks can be arbitrarily high on even out-of-distribution samples. Thus, under long-tailed scenarios where the model is generally biased towards the majority classes, the model can predict high confidence scores for the head classes even if the instances are actually from the tail classes, resulting in low precision for head classes. Given the drawbacks of using the confidence score as the pseudo-label criterion, we seek to design a better approach to determine if an unlabeled sample should be pseudo-labeled.

In this work, we present a novel approach for pseudo-labeling that addresses the drawbacks of confidence-based pseudo-labeling in imbalanced SSL. Instead of relying on a model's prediction confidence to decide whether to pseudo-label an unlabeled instance or not, we propose to view the pseudo-labeling decision as **an evolving in-distribution vs. out-of-distribution classification problem**[1]. Initially, only the ground-truth human-labeled samples are considered "in-distribution" because they are the only training examples. In each ensuing training iteration, the unlabeled samples that are close to the current "in-distribution" samples are pseudo-labeled and contribute to training, which in turn gradually expands the "in-distribution". Thus, any "out-of-distribution" unlabeled samples from previous iterations may become "in-distribution" in later iterations, as the distribution of the pseudo-labeled training data is continuously updated and expanded. An illustrative example of this process can be found in Figure 1.

To identify the "inliers", we leverage the energy score (LeCun et al., 2006) for its simplicity and good empirical performance. The energy score is a non-probabilistic scalar that is derived from a model's logits and theoretically aligned with the probability density of a data sample—lower/higher energy reflects data with higher/lower likelihood of occurrence following the training distribution, and has been shown to be useful for conventional out-of-distribution (OOD) detection (Liu et al., 2020). In our imbalanced SSL setting, at each training iteration, we compute the energy score for each unlabeled sample. If an unlabeled sample's energy is below a certain threshold, we pseudo-label it with the predicted class made by the model. To the best of the authors' knowledge, our work is the first to consider pseudo-labeling in imbalanced SSL from an *in-distribution vs. out-distribution* perspective and is also the first work that performs pseudo-labeling without using softmax scores. We refer to our method as **In**lier **P**seudo-**L**abeling (**InPL**) in the rest of this paper.

To evaluate the proposed InPL, we integrate it into the classic FixMatch (Sohn et al., 2020) framework and the recent state-of-the-art imbalanced SSL framework ABC (Lee et al., 2021) by replacing their vanilla confidence-based pseudo-labeling with our energy-based pseudo-labeling. InPL significantly

---

[1]Note that our definition of out-of-distribution is different from the typical definition from the Out-of-Distribution literature that constitutes unseen classes.

outperforms the confidence-based counterparts in both frameworks under long-tailed SSL scenarios. For example, InPL produces around 3% absolute accuracy improvement on CIFAR10-LT over vanilla FixMatch. With the ABC framework, our approach achieves a 6.9% accuracy improvement in one of the most challenging evaluation scenarios. Our analysis further shows that pseudo-labels produced using InPL achieve higher overall precision, and a significant improvement in recall for the tail classes without hurting their precision much. This result demonstrates that the pseudo-labeling process becomes more reliable with InPL in long-tailed scenarios. InPL also achieves favourable improvement on large-scaled datasets, shows a high level of robustness against the presence of real OOD examples in unlabeled data, and attains competitive performance on standard SSL benchmarks, which are (roughly) balanced.

## 2 RELATED WORK

**Class-Imbalanced Semi-Supervised Learning.** While SSL research (Scudder, 1965; McLachlan, 1975) has been extensively studied in the balanced setting in which all categories have (roughly) the same number of instances, class-imbalanced SSL is much less explored. A key challenge is to avoid overfitting to the majority classes while capturing the minority classes. State-of-the-art imbalanced SSL methods build upon recent advances in balanced SSL that combine pseudo-labeling (Lee et al., 2013) with consistency regularization (Laine & Aila, 2017; Tarvainen & Valpola, 2017; Berthelot et al., 2019; 2020; Xie et al., 2020b; Sohn et al., 2020; Xie et al., 2020a; Zhang et al., 2021; Xu et al., 2021), leveraging standard frameworks such as FixMatch (Sohn et al., 2020) to predict pseudo-labels on weakly-augmented views of unlabeled images and train the model to predict those pseudo labels on strongly-augmented views. DARP (Kim et al., 2020) refines the pseudo-labels through convex optimization targeted specifically for the imbalanced scenario. CReST (Wei et al., 2021) achieves class-rebalancing by pseudo-labeling unlabeled samples with frequency that is inversely proportional to the class frequency. Adsh (Guo & Li, 2022) extends the idea of adaptive thresholding (Zhang et al., 2021; Xu et al., 2021) to the long-tailed scenario. ABC (Lee et al., 2021) introduces an auxiliary classifier that is trained with class-balanced sampling whereas DASO (Oh et al., 2022) uses a similarity-based classifier to complement pseudo-labeling. Our method is in parallel to these developments – it can be easily integrated into these prior approaches by replacing their confidence-based thresholding with our energy-based one; for example, when plugged into ABC, InPL achieves significant performance gains. Importantly, ours is the first to view the pseudo-labeling process for imbalanced SSL from an "in-distribution vs. out-of-distribution" perspective.

**Out-of-Distribution Detection.** OOD detection aims to detect outliers that are substantially different from the training data, and is important when deploying ML models to ensure safety and reliability in real-world settings. The softmax score was used as a baseline for OOD detection in (Hendrycks & Gimpel, 2017) but has since been proven to be an unreliable measure by subsequent work (Hein et al., 2019; Liu et al., 2020). Improvements have been made in OOD detection through temperatured softmax (Liang et al., 2018) and the energy score (Liu et al., 2020). Robust SSL methods (Chen et al., 2020; Saito et al., 2021; Guo et al., 2020; Yu et al., 2020) aim to improve the model's performance when OOD examples are present in the unlabeled and/or test data under SSL scenarios. For example, D3SL (Guo et al., 2020) optimizes a meta network to selectively use unlabeled data; OpenMatch (Saito et al., 2021) trains an outlier detector with soft consistency regularization. These robust SSL methods are used as additional baselines for evaluation on imbalanced SSL in Section 4.1. Exploring new methods in OOD detection is not the focus of our work. Instead, we show that leveraging the concept of OOD detection, and in particular, the energy score (LeCun et al., 2006; Liu et al., 2020) for pseudo-labeling, provides a new perspective in imbalanced SSL that results in significant improvement under imbalanced settings.

## 3 APPROACH

Our goal is to devise a more reliable way of pseudo-labeling in imbalanced SSL, which accounts for whether an unlabeled data sample can be considered "in-distribution" or "out-of-distribution" based on the existing set of (pseudo-)labeled samples. To help explain our approach, Inlier Pseudo-Labeling (InPL), we first overview the consistency regularization with confidence-based pseudo-labeling framework (Sohn et al., 2020; Zhang et al., 2021; Xie et al., 2020a) that state-of-the-art SSL methods build upon, as our method simply replaces one step — the pseudo-labeling criterion. We then provide a detailed explanation of InPL.

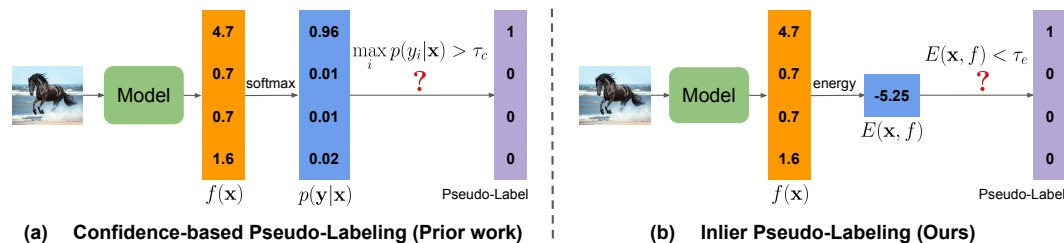

Figure 2: Overview of Confidence-based Pseudo-Labeling vs. Inlier Pseudo-Labeling.

### 3.1 BACKGROUND: CONSISTENCY REGULARIZATION WITH CONFIDENCE-BASED PSEUDO-LABELING

The training of pseudo-labeling SSL methods for image classification involves two loss terms: the supervised loss $\mathcal{L}_s$ computed on human-labeled data and the unsupervised loss $\mathcal{L}_u$ computed on unlabeled data. The supervised loss is typically the standard multi-class cross-entropy loss computed on weakly-augmented views (e.g., flip and crop) of labeled images. Let $\mathcal{X} = \{(\mathbf{x_b}, \mathbf{y_b})\}_{b=1}^{B_s}$ be the labeled set where $\mathbf{x}$ and $\mathbf{y}$ denote the data sample and its corresponding one-hot label, respectively. Denote $p(\mathbf{y}|\omega(\mathbf{x_b})) = f(\omega(\mathbf{x_b}))$ as the predicted class distribution on input $\mathbf{x_b}$, where $\omega$ is a weakly-augmenting transformation and $f$ is a classifier often realized as a deep network. Then at each iteration, the supervised loss for a batch $B_s$ of labeled data is given by

$$\mathcal{L}_s = \frac{1}{B_s} \sum_{b=1}^{B_s} \mathcal{H}(\mathbf{y_b}, p(\mathbf{y}|\omega(\mathbf{x_b}))), \tag{1}$$

where $\mathcal{H}$ is the cross-entropy loss.

Mainstream research in SSL focuses on how to construct the unsupervised loss. One dominating approach is consistency regularization (Laine & Aila, 2017; Tarvainen & Valpola, 2017; Berthelot et al., 2019; 2020), which regularizes the network to be less sensitive to input or model perturbations by enforcing consistent predictions across different views (augmentations) of the same training input, through an MSE or KL-divergence loss. Self-training (Lee et al., 2013; Rosenberg et al., 2005; Rizve et al., 2021; Xie et al., 2020b) converts model predictions to optimization targets for unlabeled images. In particular, pseudo-labeling (Lee et al., 2013) converts model predictions into hard pseudo-labels. To ensure high quality of pseudo-labels, a high confidence threshold is often used.

The recently introduced weak-strong data augmentation paradigm (Sohn et al., 2020; Xie et al., 2020a) can be viewed as the combination of these two directions. When combined with confidence-based pseudo-labeling (Sohn et al., 2020; Zhang et al., 2021), at each iteration, the process can be summarized as follows:

1. For each unlabeled data point $\mathbf{x}$, the model makes prediction $p(\mathbf{y}|\omega(\mathbf{x})) = f(\omega(\mathbf{x_b}))$ on its weakly-augmented view $\omega(\mathbf{x})$.

2. Confidence thresholding is applied and a pseudo-label is only produced when the maximum predicted probability $\max_i p(y_i|\omega(\mathbf{x}))$ of $\mathbf{x}$ is above a threshold $\tau_c$ (typically, $\tau_c = 0.95$).

3. The model is then trained with its strongly-augmented view $\Omega(\mathbf{x})$ (e.g., RandAugment (Cubuk et al., 2020) and CutOut (DeVries & Taylor, 2017)) along with its one-hot thresholded pseudo-label $\hat{p}(\mathbf{y}|\omega(\mathbf{x}))$ obtained on the weakly-augmented view.

With batch size $B_u$ for unlabeled data, the unsupervised loss is formulated as follows:

$$\mathcal{L}_u = \frac{1}{B_u} \sum_{b=1}^{B_u} \mathbb{1}[\max_i(p(y_i|\omega(\mathbf{x_b}))) \geq \tau_c] \, \mathcal{H}(\hat{p}(\mathbf{y}|\omega(\mathbf{x_b})), p(\mathbf{y}|\Omega(\mathbf{x_b}))), \tag{2}$$

where $\mathbb{1}[\cdot]$ is the indicator function.

The final loss at each training iteration is computed by $\mathcal{L} = \mathcal{L}_s + \lambda \mathcal{L}_u$ with $\lambda$ as a hyperparameter to balance the loss terms. The model parameters are updated with this loss after each iteration.

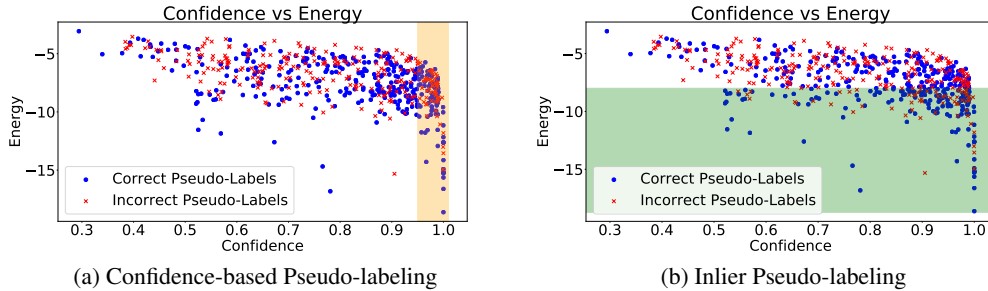

Figure 3: **Visualization: confidence vs energy score**: The shaded region shows the unlabeled samples that are pseudo-labeled. Inlier Pseudo-Labeling can produce correct pseudo-labels for many low-confident unlabeled samples, increasing recall while filtering out many false positives.

## 3.2 CONSISTENCY REGULARIZATION WITH INLIER PSEUDO-LABELING

Confidence-based thresholding (step 2 in the above process) typically produces pseudo-labels with high precision in the balanced SSL setting, yet often leads to low precision for head classes and low recall for tail classes in the long-tailed scenario (Wei et al., 2021). Moreover, softmax-based confidence scores in deep networks are oftentimes overconfident (Guo et al., 2017), and can be arbitrarily high for samples that are far from the training data (Nguyen et al., 2015; Hein et al., 2019). The implication in the imbalanced SSL setting is that the pseudo-label for even high-confidence unlabeled samples may not be trustworthy if those samples are far from the labeled data.

To address these issues, our method tackles the imbalanced SSL pseudo-labeling process from a different perspective: instead of generating pseudo-labels for high-confidence samples, we produce pseudo-labels only for unlabeled samples that are close to the current training distribution — we call these "in-distribution" samples. The rest are "out-of-distribution" samples, for which the model's confidences are deemed unreliable. The idea is that, as training progresses and more unlabeled samples become in-distribution and contribute to training, the training distribution will better approximate the true distribution to improve the model, and in turn, improve the overall reliability of the pseudo-labels (see Figure 1).

To determine whether an unlabeled sample is in-distribution or out-of-distribution, we use the energy score (LeCun et al., 2006) derived from the classifier $f$. The energy score is defined as:

$$E(\mathbf{x}, f(\mathbf{x})) = -T \cdot \log(\sum_{i=1}^{K} e^{f_i(\mathbf{x})/T}), \tag{3}$$

where $\mathbf{x}$ is the input data and $f_i(\mathbf{x})$ indicates the corresponding logit value of the $i$-th class. $K$ is the total number of classes and $T$ is a tunable temperature.

When used for conventional OOD detection, smaller/higher energy scores indicate that the input is likely to be in-distribution/out-of-distribution. Indeed, a discriminative classifier implicitly defines a free energy function (Grathwohl et al., 2020; Xie et al., 2016) that can be used to characterize the data distribution (LeCun et al., 2006; Liu et al., 2020). This is because the training of the classifier, when using the negative log-likelihood loss, seeks to minimize the energy of in-distribution data points. Since deriving and analyzing the energy function is not the focus of our paper, we refer the reader to Liu et al. (2020) for a detailed connection between the energy function and OOD detection.

In our Inlier Pseudo-Labeling framework, we compute the energy score for each unlabeled sample and only generate a pseudo-label when the corresponding energy score is less than a pre-defined threshold $\tau_e$, which indicates that the unlabeled sample is close to the current training distribution. The actual pseudo-label is obtained by converting the model prediction on the weakly-augmented view of $\omega(\mathbf{x_b})$ to a one-hot pseudo-label. Formally, the unsupervised loss is defined as:

$$\mathcal{L}_u = \frac{1}{B_u} \sum_{b=1}^{B_u} \mathbb{1}[E(\omega(\mathbf{x_b}), f(\omega(\mathbf{x_b}))) < \tau_e] \, \mathcal{H}(\hat{p}(\mathbf{y}|\omega(\mathbf{x_b})), p(\mathbf{y}|\Omega(\mathbf{x_b}))). \tag{4}$$

We illustrate the key difference between confidence-based pseudo-labeling and energy-based pseudo-labeling in Figure 2. It is worth noting that the energy score is not the only possible metric to implement our InPL approach and it is possible to use other OOD detection metrics for the objective.

| | CIFAR10-LT | | | CIFAR100-LT | |
|---|---|---|---|---|---|
| | $\gamma = 50$ | $\gamma = 100$ | $\gamma = 200$ | $\gamma = 50$ | $\gamma = 100$ |
| UDA (Xie et al., 2020a) | $80.21_{\pm0.49}$ | $72.19_{\pm1.51}$ | $63.32_{\pm1.67}$ | $46.79_{\pm0.76}$ | $41.47_{\pm0.97}$ |
| FixMatch (Sohn et al., 2020) | $80.84_{\pm0.20}$ | $72.95_{\pm1.32}$ | $63.25_{\pm0.13}$ | $46.99_{\pm0.37}$ | $41.49_{\pm0.38}$ |
| FixMatch-UPS (Rizve et al., 2021) | $81.75_{\pm0.56}$ | $73.17_{\pm1.63}$ | $64.38_{\pm0.56}$ | - | - |
| FixMatch-InPL w/o AML (ours) | $\mathbf{83.36}_{\pm0.38}$ | $\mathbf{76.05}_{\pm0.84}$ | $\mathbf{66.47}_{\pm1.06}$ | $\mathbf{48.03}_{\pm0.31}$ | $\mathbf{42.53}_{\pm0.68}$ |
| FixMatch-Debias + AML (Wang et al., 2022) | $83.53_{\pm0.67}$ | $76.92_{\pm1.72}$ | $67.70_{\pm0.44}$ | $\mathbf{50.24}_{\pm0.46}$ | $44.12_{\pm0.81}$ |
| FixMatch-InPL(ours) | $\mathbf{83.92}_{\pm0.52}$ | $\mathbf{77.44}_{\pm1.17}$ | $\mathbf{68.47}_{\pm1.15}$ | $49.96_{\pm0.36}$ | $\mathbf{44.33}_{\pm0.61}$ |
| OpenMatch (Saito et al., 2021) | $81.01_{\pm0.45}$ | $73.15_{\pm1.03}$ | $63.22_{\pm1.86}$ | $46.92_{\pm0.28}$ | $40.76_{\pm0.81}$ |
| FixMatch-D3SL (Guo et al., 2020) | $81.20_{\pm0.33}$ | $72.71_{\pm2.32}$ | $65.09_{\pm1.72}$ | $46.83_{\pm0.45}$ | $41.22_{\pm0.39}$ |

Table 1: **Top-1 accuracy of FixMatch variants on CIFAR 10-LT/100-LT**. For CIFAR10-LT and CIFAR100-LT, we use 10% and 30% data as labeled sets, respectively. We use Wide ResNet-28-2 (Zagoruyko & Komodakis, 2016) for CIFAR 10-LT and WRN-28-8 for CIFAR 100-LT. All methods are trained with the default FixMatch training schedule (Sohn et al., 2020). Results are reported with the mean and standard deviation over 3 different runs.

We choose the energy score for its simplicity and easy integration to existing SSL frameworks such as FixMatch and ABC, which does not incur additional model parameters or significant computation cost (apart from the computation of the energy score).

To further tackle the model bias in long-tail scenarios, we can also optionally replace the vanilla cross-entropy loss ($\mathcal{H}(\cdot)$) in the unsupervised loss computation with a margin-aware loss (Li et al., 2019; Khan et al., 2019; Cao et al., 2019; Wang et al., 2022). We adopt the adaptive margin loss (Wang et al., 2022) with the margin determined through the moving average of the model's prediction:

$$\mathcal{L}_{AML} = -\log \frac{e^{f_i(\Omega(\mathbf{x_b})) - \Delta_i}}{e^{f_i(\Omega(\mathbf{x_b})) - \Delta_i} + \sum_{k \neq i}^{C} e^{f_k(\Omega(\mathbf{x_b})) - \Delta_k}}, \tag{5}$$

where $f_i(\Omega(\mathbf{x_b}))$ represents the corresponding logits of class $i$ on strongly-augmented input $\Omega(\mathbf{x_b})$. The margin is computed by $\Delta_i = \lambda \log(\frac{1}{\tilde{p}_i})$, where $\tilde{p}$ is the averaged model prediction updated at each iteration through an exponential moving average. We shall see in the experiments that InPL outperforms the vanilla confidence-based counterparts regardless of whether we use the vanilla cross-entropy loss or the adaptive margin loss.

## 4 EXPERIMENTS

In this section, we first compare our energy-based pseudo-labeling approach to confidence-based pseudo-labeling approaches that build upon FixMatch (Sohn et al., 2020) and ABC (Lee et al., 2021) framework and compare to state-of-the-art imbalanced SSL approaches. We evaluate our method on both standard imbalanced SSL benchmarks and the large-scale ImageNet dataset. Next, we provide analyses on why InPL works well in the imbalanced setting, and then evaluate InPL's robustness to real OOD samples in the unlabeled data. All experiments are conducted with multiple runs using different random seeds and we report the mean and standard deviation of the final accuracy. Following prior suggestion (Oliver et al., 2018) on SSL, we implement our method and baseline methods in the same codebase. Unless otherwise specified, results for each method are generated with the same codebase, same random seeds, same data splits, and same network architecture for the best possible fair comparison. Implementation details of all experiments can be found in Appendix A.2.

### 4.1 ENERGY-BASED PSEUDO-LABELING VS. CONFIDENCE-BASED PSEUDO LABELING

We first evaluate the effectiveness of InPL's energy-based pseudo labeling over standard confidence-based pseudo labeling. For this, we integrate InPL into the FixMatch framework (Sohn et al., 2020) (denoted as FixMatch-InPL), and compare it with the following FixMatch variants, which all use confidence-based pseudo-labeling: UDA (Xie et al., 2020a), FixMatch-Debiased (Wang et al., 2022), and UPS (Rizve et al., 2021). UDA uses soft pseudo-labels with temperature sharpening; FixMatch-Debiased (Wang et al., 2022) adds a debiasing module before computing the confidence score in

| Dataset | CIFAR10-LT | | | CIFAR100-LT |
|---|---|---|---|---|
| Imbalance Ratio | $\gamma = 100$ | $\gamma = 150$ | $\gamma = 200$ | $\gamma = 20$ |
| FixMatch (Sohn et al., 2020) | $72.3_{\pm 0.33}$ / $53.8_{\pm 0.63}$ | $68.5_{\pm 0.60}$ / $45.8_{\pm 1.15}$ | $66.3_{\pm 0.49}$ / $42.4_{\pm 0.94}$ | $51.0_{\pm 0.20}$ / $32.8_{\pm 0.41}$ |
| w/ DARP+cRT (Kim et al., 2020) | $78.1_{\pm 0.89}$ / $66.6_{\pm 1.55}$ | $73.2_{\pm 0.85}$ / $57.1_{\pm 1.13}$ | - | $54.7_{\pm 0.46}$ / $41.2_{\pm 0.42}$ |
| w/ CReST+ (Wei et al., 2021) | $76.6_{\pm 0.46}$ / $61.4_{\pm 0.85}$ | $70.0_{\pm 0.82}$ / $49.4_{\pm 1.52}$ | - | $51.6_{\pm 0.29}$ / $36.4_{\pm 0.46}$ |
| w/ ABC (Lee et al., 2021) | $81.1_{\pm 0.82}$ / $72.0_{\pm 1.77}$ | $77.1_{\pm 0.46}$ / $64.4_{\pm 0.92}$ | $73.9_{\pm 1.18}$ / $58.1_{\pm 2.72}$ | $56.3_{\pm 0.19}$ / $43.4_{\pm 0.42}$ |
| w/ ABC-InPL (ours) | $\mathbf{82.9}_{\pm 0.60}$ / $\mathbf{76.4}_{\pm 1.49}$ | $\mathbf{79.7}_{\pm 0.71}$ / $\mathbf{70.8}_{\pm 1.43}$ | $\mathbf{76.4}_{\pm 1.09}$ / $\mathbf{63.7}_{\pm 2.03}$ | $\mathbf{57.7}_{\pm 0.33}$ / $\mathbf{46.4}_{\pm 0.26}$ |
| RemixMatch (Berthelot et al., 2020) | $73.7_{\pm 0.39}$ / $55.9_{\pm 0.87}$ | $69.9_{\pm 0.23}$ / $48.4_{\pm 0.60}$ | $68.2_{\pm 0.37}$ / $45.4_{\pm 0.70}$ | $54.0_{\pm 0.29}$ / $37.1_{\pm 0.37}$ |
| w/ DARP+cRT (Kim et al., 2020) | $78.5_{\pm 0.61}$ / $66.4_{\pm 1.69}$ | $73.9_{\pm 0.59}$ / $57.4_{\pm 1.45}$ | - | $55.1_{\pm 0.45}$ / $43.6_{\pm 0.58}$ |
| w/ CReST+ (Wei et al., 2021) | $75.7_{\pm 0.34}$ / $59.6_{\pm 0.76}$ | $71.3_{\pm 0.77}$ / $50.8_{\pm 1.56}$ | - | $54.6_{\pm 0.48}$ / $38.1_{\pm 0.69}$ |
| w/ ABC (Lee et al., 2021) | $82.4_{\pm 0.45}$ / $75.7_{\pm 1.18}$ | $80.6_{\pm 0.66}$ / $72.1_{\pm 1.51}$ | $78.8_{\pm 0.27}$ / $69.9_{\pm 0.99}$ | $57.6_{\pm 0.26}$ / $46.7_{\pm 0.50}$ |
| w/ ABC-InPL(ours) | $\mathbf{83.6}_{\pm 0.45}$ / $\mathbf{81.7}_{\pm 0.97}$ | $\mathbf{81.3}_{\pm 0.83}$ / $\mathbf{76.8}_{\pm 0.88}$ | $\mathbf{78.8}_{\pm 0.75}$ / $\mathbf{74.5}_{\pm 1.47}$ | $\mathbf{58.4}_{\pm 0.25}$ / $\mathbf{48.9}_{\pm 0.36}$ |

Table 2: **Top-1 accuracy on long-tailed CIFAR10/100 following ABC (Lee et al., 2021) evaluations.** We use 20% labeled data for CIFAR10-LT and 40% labeled data for CIFAR100-LT. We report both the overall accuracy (before "/") and the accuracy of minority classes (after "/").

pseudo-labeling; UPS (Rizve et al., 2021) uses an uncertainty estimation with MC-Dropout (Gal & Ghahramani, 2016) as an extra criteria in pseudo-labeling. We also compare with recent robust SSL methods, OpenMatch (Saito et al., 2021) and D3SL (Guo et al., 2020), as they connect SSL to OOD detection. Since UPS and D3SL do not use strong augmentations, for fair comparison, we integrate them into the FixMatch framework, denoted by FixMatch-UPS and FixMatch-D3SL. We evaluate on CIFAR10-LT and CIFAR100-LT (Krizhevsky & Hinton, 2009), which are long-tailed variants of the original CIFAR datasets. We follow prior work in long-tail SSL (Wei et al., 2021; Lee et al., 2021), and use an exponential imbalance function (Cui et al., 2019) to create the long-tailed version of CIFAR10 and CIFAR100. Details of constructing these datasets are in Appendix A.1.

**InPL outperforms confidence-based pseudo-labeling approaches for imbalanced SSL**. Our FixMatch-InPL shows a significant improvement over FixMatch variants with confidence-based pseudo-labeling by a large margin (e.g., 3% absolute percentage over FixMatch for CIFAR10-LT); see Table 1. For CIFAR100-LT, InPL reaches 48.03% and 42.53% average accuracy when $\gamma = 50$ and $\gamma = 100$ respectively, outperforming other methods. UPS is the closest competitor as it tries to measure the uncertainty in the pseudo-labels. Using UPS in FixMatch improves the performance over vanilla FixMatch, yet it still cannot match the performance of InPL on CIFAR10-LT. Moreover, UPS requires forwarding the unlabeled data 10 times to compute its uncertainty measurement, which is extremely expensive in modern SSL frameworks such as FixMatch. In comparison to FixMatch-UPS, our InPL provides strong empirical results on long-tailed data and remains highly efficient. Comparing InPL and FixMatch-debiased (both use AML), our approach again shows consistent improvement on CIFAR10-LT. We also outperform the robust SSL methods, OpenMatch and D3SL. Overall, these results show the efficacy of InPL over standard confidence-based pseudo-labeling for imbalanced SSL.

## 4.2 COMPARISON TO STATE-OF-THE-ART IMBALANCED SSL APPROACHES

Next, we compare InPL to state-of-the-art imbalanced SSL approaches. We integrate InPL into ABC (Lee et al., 2021), a recent framework designed for imbalanced SSL (denoted as ABC-InPL), by replacing its confidence-based pseudo-labeling with our energy-based pseudo-labeling. We compare our ABC-InPL with vanilla ABC as well as other state-of-the-art imbalanced SSL methods, DARP (Kim et al., 2020), CREST (Wei et al., 2021), Adsh (Guo & Li, 2022) and DASO (Oh et al., 2022). The results are shown in Tables 2 and 3, which correspond to different ways of constructing the imbalanced SSL datasets. See implementations details in Appendix A.1 and A.2.

**InPL improves the state-of-the-art for imbalanced SSL**. Table 2 shows that ABC-InPL consistently outperforms the confidence-based counterparts on CIFAR10-LT and CIFAR100-LT under the ABC framework. For FixMatch base, when the imbalance ratio $\gamma = 150$ and $\gamma = 200$ on CIFAR10-LT, InPL shows a 2.6% and 2.5% improvement, respectively (FixMatch w/ ABC vs. FixMatch w/ ABC-InPL). Further, we demonstrate the flexibility of InPL by incorporating it into RemixMatch, a consistency-based SSL method that uses Mixup (Zhang et al., 2018). With RemixMatch, InPL again consistently outperforms confidence-based methods. Importantly, ABC-InPL achieves significantly higher accuracy for minority classes across all experiment protocols (numbers reported after the "/"

| | CIFAR10-LT | | | | CIFAR100-LT | | | |
| --- | --- | --- | --- | --- | --- | --- | --- | --- |
| | $\gamma = 100$ | | $\gamma = 150$ | | $\gamma = 10$ | | $\gamma = 20$ | |
| | $N_1 = 500$ $M_1 = 4000$ | $N_1 = 1500$ $M_1 = 3000$ | $N_1 = 500$ $M_1 = 4000$ | $N_1 = 1500$ $M_1 = 3000$ | $N_1 = 50$ $M_1 = 400$ | $N_1 = 150$ $M_1 = 300$ | $N_1 = 50$ $M_1 = 400$ | $N_1 = 150$ $M_1 = 300$ |
| FixMatch† (Sohn et al., 2020) | $68.5_{\pm 0.94}$ | $71.5_{\pm 0.31}$ | $62.9_{\pm 0.36}$ | $72.4_{\pm 1.03}$ | - | - | - | - |
| w/ Adsh† (Cui et al., 2019) | $76.3_{\pm 0.86}$ | $78.1_{\pm 0.42}$ | $67.5_{\pm 0.45}$ | $73.7_{\pm 0.34}$ | - | - | - | - |
| FixMatch (Sohn et al., 2020) | $67.8_{\pm 1.13}$ | $77.5_{\pm 1.32}$ | $62.9_{\pm 0.36}$ | $72.4_{\pm 1.03}$ | $45.2_{\pm 0.55}$ | $56.5_{\pm 0.06}$ | $40.0_{\pm 0.96}$ | $50.7_{\pm 0.25}$ |
| w/ DARP (Kim et al., 2020) | $74.5_{\pm 0.78}$ | $77.8_{\pm 0.63}$ | $67.2_{\pm 0.32}$ | $73.6_{\pm 0.73}$ | $49.4_{\pm 0.20}$ | $58.1_{\pm 0.44}$ | $43.4_{\pm 0.87}$ | $52.2_{\pm 0.66}$ |
| w/ CREST+ (Wei et al., 2021) | $76.3_{\pm 0.86}$ | $78.1_{\pm 0.42}$ | $67.5_{\pm 0.45}$ | $73.7_{\pm 0.34}$ | $44.5_{\pm 0.94}$ | $57.4_{\pm 0.18}$ | $40.1_{\pm 1.28}$ | $50.1_{\pm 0.21}$ |
| w/ DASO (Oh et al., 2022) | $76.0_{\pm 0.37}$ | $79.1_{\pm 0.75}$ | $70.1_{\pm 1.81}$ | $75.1_{\pm 0.77}$ | $49.8_{\pm 0.24}$ | $59.2_{\pm 0.35}$ | $43.6_{\pm 0.09}$ | $52.9_{\pm 0.42}$ |
| w/ ABC (Lee et al., 2021) | $78.9_{\pm 0.82}$ | $83.8_{\pm 0.36}$ | $66.5_{\pm 0.78}$ | $80.1_{\pm 0.45}$ | $47.5_{\pm 0.18}$ | $59.1_{\pm 0.21}$ | $41.6_{\pm 0.83}$ | $53.7_{\pm 0.55}$ |
| w/ ABC-DASO (Oh et al., 2022) | $80.1_{\pm 1.16}$ | $83.4_{\pm 0.31}$ | $70.6_{\pm 0.80}$ | $80.4_{\pm 0.56}$ | $50.2_{\pm 0.62}$ | $60.0_{\pm 0.32}$ | $44.5_{\pm 0.25}$ | $\mathbf{55.3}_{\pm 0.53}$ |
| w/ ABC-InPL (Ours) | $\mathbf{81.4}_{\pm 0.76}$ | $\mathbf{84.4}_{\pm 0.20}$ | $\mathbf{77.5}_{\pm 1.57}$ | $\mathbf{80.9}_{\pm 0.82}$ | $\mathbf{51.8}_{\pm 1.09}$ | $\mathbf{61.0}_{\pm 0.32}$ | $\mathbf{44.6}_{\pm 1.24}$ | $55.1_{\pm 0.51}$ |

Table 3: **Top-1 accuracy on long-tailed CIFAR10/100 compared with SSL-LT methods following DASO (Oh et al., 2022) evaluations**. $N_1$ and $M_1$ represent the number of instances from the most-majority class. †We use the reported results in Adsh (Cui et al., 2019) due to adaptation difficulties.

in Table 2). This indicates that our "in-distribution vs out-of-distribution" perspective leads to more reliable pseudo-labels when labeled data is scarce, compared to confidence-based selection.

Table 3 shows more results under different combinations of labeled instances and imbalance ratios. Across almost all settings, InPL consistently outperforms the baselines, sometimes by a very large margin. For example, in the most challenging scenario ($\gamma = 150$ and $N_1 = 500$), ABC-InPL achieves 77.5% absolute accuracy, which outperforms the state-of-the-art ABC-DASO baseline by 6.9% absolute accuracy. These results demonstrate that InPL achieves strong performance on imbalanced data across different frameworks and evaluation settings.

## 4.3 RESULTS ON IMAGENET

We evaluate InPL on ImageNet-127 (Huh et al., 2016), a large-scale dataset where the 1000 ImageNet classes are grouped into 127 classes based on the WordNet hierarchy. This introduces a class imbalance with ratio 286. Due to limited computation, we are unable to use large batch sizes as in FixMatch (Sohn et al., 2020) (1024 for labeled data and 5120 for unlabeled data). Instead, we use a batch size of 128 for both labeled and unlabeled data following the

| | ImageNet-127 (Imbalanced) | ImageNet (Balanced) |
| --- | --- | --- |
| FixMatch | 51.96 | 56.34 |
| FixMatch-InPL (ours) | **54.82** | **57.92** |

Table 4: Results on ImageNet-127 and ImageNet. We use sample 10% data as the labeled set for ImageNet-127 and use 100 labels per class for ImageNet. Our approach outperforms the confidence-based counterpart in FixMatch on both datasets.

recent SSL benchmark (Zhang et al., 2021). As shown in Table 4, InPL outperforms the vanilla FixMatch with confidence-based pseudo-labeling by a large margin, which further demonstrates the efficacy of our approach on large-scaled datasets.

While designed to overcome the limitations of confidence-based pseudo-labeling on imbalanced data, InPL does not make any explicit assumptions about the class distribution. Thus, a natural question is: how does InPL perform on standard SSL benchmarks? To answer this, we further evaluate InPL on ImageNet with standard SSL settings in Table 4 (results on other standard SSL datasets can be found in Appendix A.8) and InPL still shows favourable improvement. This shows that the energy-based pseudo-labeling also has potential to become a general solution to SSL problems.

## 4.4 WHY DOES INPL WORK WELL ON IMBALANCED DATA?

To help explain the strong performance of InPL on imbalanced data, we provide a detailed pseudo-label precision and recall analysis on CIFAR10-LT. Here, we refer to the three most frequent classes as head classes, the three least frequent classes as tail classes, and the rest as body classes.

Figure 4 shows the precision and recall of our model's predicted pseudo-labels over all classes (a,c) and also for the tail classes (b,d). The analysis for the head and body classes can be found in Appendix A.4. Compared with FixMatch, InPL achieves higher precision for overall, head, and body pseudo-labels. Importantly, it doubles FixMatch's recall of tail pseudo-labels without hurting

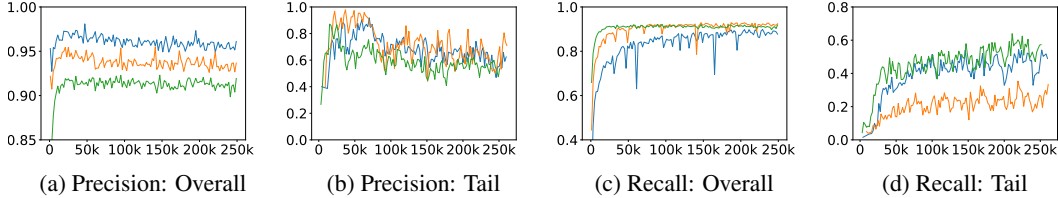

|  |  |  |  |
|---|---|---|---|
| (a) Precision: Overall | (b) Precision: Tail | (c) Recall: Overall | (d) Recall: Tail |

Figure 4: **Precision-Recall Analysis**: We compare pseudo-label precision and recall between InPL and FixMatch. Orange and green curves denote FixMatch with threshold 0.95 and 0.6 respectively. InPL is shown in blue, which achieves improved recall for tail classes and better overall precision.

the precision much. This shows that InPL predicts more true positives for the tail classes and also becomes less biased to the head classes. Trivially lowering the confidence threshold for FixMatch can also improve the recall for tail pseudo-labels. However, doing so significantly hurts its precision and does not improve overall accuracy. For example, as shown in Figure 4 (a) and (d), although FixMatch with a lower threshold of 0.6 achieves higher recall for tail pseudo-labels, the overall precision is significantly hurt, which results in minimal improvement or degradation in overall accuracy. We further investigate various lower confidence thresholds for FixMatch and find that none of them leads to improvement in accuracy to match the performance level of InPL (see Appendix A.5).

### 4.5 Additional Analyses

**InPL is robust against OOD examples**. In real-world scenarios, unlabeled data may not share the same set of classes as labeled data. To simulate this, we evaluate our method with out-of-distribution data presenting in the unlabeled data. Here, we consider the scenario where unlabeled data contain examples from different datasets. We sample 4 labeled examples per class from CIFAR10 and use the rest of CIFAR10 and SVHN as the unlabeled data. Another OOD robustness experiment under the scenario of class mismatch in unlabeled data (Oliver et al., 2018) can be found in Appendix A.6.

Although motivated by imbalanced SSL, InPL also achieves strong performance under the aforementioned realistic conditions where unlabeled instances from different data domains are present in the unlabeled set. We compare InPL with the confidence-based methods UDA (Xie et al., 2020a) and FixMatch (Sohn et al., 2020). When a large amount of OOD examples are in the unlabeled set, the overall performance of both methods decreases, but InPL demonstrates a significant advantage. Specifically, InPL outperforms UDA and FixMatch by 4.21% and 4.42% absolute accuracy, respectively. Moreover, InPL consistently includes less OOD examples in training (see Appendix A.6 for details). This experiment shows the robustness of InPL to true outliers.

|  | Accuracy |
|---|---|
| UDA | 62.81 |
| FixMatch | 62.60 |
| FixMatch-InPL (ours) | **67.02** |

Table 5: Results when OOD samples from SVHN appear in the unlabeled set when training a model on CIFAR10 classification.

**Other ablation studies.** Additional ablation studies can be found in the Appendix including the choice of thresholds and temperatures, impact of adaptive margin loss, and more analysis.

## 5 Discussion and Conclusion

In this work, we presented a novel "in-distribution vs. out-distribution" perspective for pseudo-labeling in imbalanced SSL, in tandem with our energy-based pseudo-labeling method (InPL). Importantly, our method selects unlabeled samples for pseudo-labeling based on their energy scores derived from the model's output. We showed that our method can be easily integrated into state-of-the-art imbalanced SSL approaches, and achieve strong results. We further demonstrated that our method renders robustness against out-of-distribution samples, and remains competitive on balanced SSL benchmarks. One limitation is the lack of interpretability of the energy scores; the energy score has a different scale and is harder to interpret. Devising ways to better understand it would be interesting future work. Overall, we believe our work has shown the promise of energy-based approaches for imbalanced SSL, and hope that it will spur further research in this direction.

**Acknowledgements.** This work was also supported in part by NSF CAREER IIS2150012, Wisconsin Alumni Research Foundation, and NASA 80NSSC21K0295.

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

## A    APPENDIX

This document complements the main paper by describing: (1) the construction of long-tailed datasets for semi-supervised learning (Appendix A.1); (2) training details of each experiment in the main paper (Appendix A.2); (3) ablation studies on choices of hyper-parameters (Appendix A.3); (4) more precision and recall analysis for pseudo-labels (Appendix A.4); (5) top-1 accuracy of FixMatch with different thresholds (Appendix A.5); (6) more results on realistic evaluations where real OOD examples are presented in the unlabeled sets (Appendix A.6); (7) contribution of adaptive margin loss in InPL (Appendix A.7); (8) additional results on standard SSL benchmarks (Appendix A.8); (9) theoretical comparison between confidence score and energy score.

For sections, figures, tables, and equations, we use numbers (e.g., Table 1) to refer to the main paper and capital letters (e.g., Table A) to refer to this supplement.

### A.1    CONSTRUCTION OF LONG-TAILED CIFAR

In this section, we describe the construction of the long-tailed version of CIFAR10/100 for semi-supervised learning. For experiments in Table 3, we follow prior work (Kim et al., 2020; Oh et al., 2022) and first set $N_1$ and $M_1$ to be the number of instances for the most frequent class in labeled sets and unlabeled sets, respectively. Next, with imbalance ratio $\gamma$, the number of instances for class $k$ in labeled sets is computed as $N_k = N_1 \cdot \gamma^{-(k-1)/(K-1)}$, where $K$ is the total number of classes and $N_1$ is the number of labels for the most frequent class. Likewise, the number of instances in unlabeled sets is computed as $M_k = M_1 \cdot \gamma^{-(k-1)/(K-1)}$.

For experiments in Table 2, we follow the construction strategy in ABC (Lee et al., 2021), which is similar to the aforementioned strategy except that $M_1 = D - N_1$ where D is the total number of instances of each class. For CIFAR10-LT and CIFAR100-LT, D is 5000 and 500 , respectively. For the experiments in Table 2, 20% labeled data for CIFAR10-LT corresponds to $N_1 = 1000$ and $M_1 = 4000$. 40% labeled data for CIFAR100-LT corresponds to $N_1 = 200$ and $M_1 = 300$.

### A.2    TRAINING DETAILS OF LONG-TAIL EXPERIMENTS

For results in Table 1, we follow the default training settings of FixMatch (Sohn et al., 2020), yet reduce the number of iterations to $6 \times 2^{16}$ following CREST (Wei et al., 2021). This is because models already converge within the reduced schedule. All other settings are exactly the same as FixMatch. Please refer to the original paper for more details.

For results in Table 2, we follow the training settings of ABC (Lee et al., 2021). Specifically, for all the results, we use Adam (Kingma & Ba, 2014) as the optimizer with a constant learning rate 0.002 and train models with 25000 iterations. Exponential moving average of momentum 0.999 is used to maintain an ensemble network for evaluation. The strong augmentation transforms include RandAugment and CutOut, which is consistent with the standard practice in SSL. Batch size of both labeled set and unlabeled set is set to 64 and two strongly-augmented views of each unlabeled sample are included in training. Following ABC, WideResNet-28-2 is used for both CIFAR10-LT and CIFAR100-LT for results in Table 2. Results are reported as the mean of five runs with different random seeds and the best accuracy of each run is recorded.

For results in Table 3, we follow the training settings of DASO (Oh et al., 2022). Most of the settings are the same as ABC (Lee et al., 2021) except that SGD is used as the optimizer. The base learning rate is set to 0.03 and evaluation is conducted every 500 iterations with total number of training iterations being 250000. The median value in last 20 evaluations of each single run is recorded and the results are reported as the mean of recorded values over three different runs.

Note that when integrating InPL into the ABC framework, our energy-based pseudo-labeling is only applied to the auxiliary class-balanced classifier. The vanilla classifier is still trained with confidence-based pseudo-labeling because empirically we find no benefit of using energy-based pseudo-labeling for both.

For all of our experiments, we simply set the weight of unsupervised loss to 1.0 without further hyper-parameter tuning.

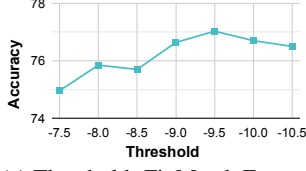 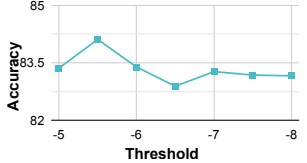 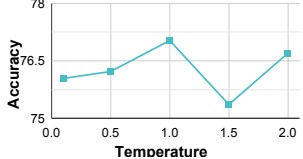

(a) Threshold: FixMatch Framework on CIFAR10-LT     (b) Threshold: ABC Framework on CIFAR10-LT     (c) Temperature: FixMatch Framework on CIFAR10-LT

Figure A: Ablation study: (a) and (b): Effects of different energy thresholds on FixMatch framework and ABC framework under CIFAR10-LT with imbalance ratio 100. For ABC framework, the ReMixmatch base is used. (c): Effects of the temperature parameter in the energy function on CIFAR10-LT under FixMatch framework.

### A.3    ABLATION STUDIES ON HYPERPARAMETERS

We conduct ablation studies to better understand our approach InPL, where we integrate it into the framework of FixMatch (Sohn et al., 2020) and ABC (Lee et al., 2021). Unless otherwise noted, experiments are conducted on CIFAR10-LT with imbalance ratio 100. For FixMatch framework, we use 10% of labeled data and for ABC framework, we use 20% of labeled data.

**Effect of energy threshold.** The most important hyperparameter of our method is the energy threshold. Unlike confidence scores that range from 0 to 1, energy scores are unbounded with their scale proportional to the number of classes. Our thresholds in the experiments are chosen via cross-validation with a separate sampling seed. We further experiment with different thresholds.

As shown in Figure A (a), within the FixMatch Framework, we observe a steady increase in the performance when the threshold goes from -7.5 to -9.5 and starts to decrease with even lower thresholds. This is because, with very low thresholds, the model becomes very conservative and only produces pseudo-labels for unlabeled samples that are very close to the training distribution; i.e., the recall in pseudo-labels takes a significant hit. As for the ABC framework (Figure A (b)), our method is not very sensitive to the energy threshold. Most threshold values result in good performance but a slightly higher threshold of -5.5 achieves the best accuracy. Since ABC already performs balanced sampling, using a higher energy threshold helps improve its recall of pseudo-labels without making the model more biased.

**Effect of temperature for the energy score.** Recall that the energy function (LeCun et al., 2006) has a tunable temperature $T$ (Equation 3). Here, we empirically evaluate the impact of this parameter. As shown in Figure A (c), the simplest setting of $T = 1$ gives the best performance. We also experiment with a much larger temperature $T = 10$, which leads to major drop in performance to 72.72% accuracy (not shown in the plot). As noted by prior work (Liu et al., 2020), larger $T$ results in a smoother distribution of energy scores, which makes it harder to distinguish samples. We find that $T < 1$ also results in slightly worse performance. Therefore, we simply set $T = 1$ and omit the temperature parameter to reduce the effort of hyperparameter tuning for our method.

### A.4    PRECISION AND RECALL FOR HEAD AND BODY PSEUDO-LABELS

Our ablation study, presented in Figure 4 of the main paper, compared the precision and recall of pseudo-labels for all classes and the tail classes. In this section, we further report the results (precision and recall of pseudo-labels) for the head and body classes. As shown in Figure B, pseudo-labels produced by InPL consistently achieve higher precision with slightly lower recall across head and body classes. This further suggests that the model trained with InPL is less biased towards the frequent classes in comparison to confidence-based pseudo-labeling.

### A.5    ACCURACY OF FIXMATCH WITH DIFFERENT THRESHOLDS

As was shown in Section 4.4, trivially lowering the confidence threshold significantly hurts the pseudo-label precision. We further demonstrate the corresponding overall accuracy in Table A. We see that none of these confidence thresholds for FixMatch is able to match the performance of our approach InPL in the long-tailed scenario, which further shows the effectiveness of our approach.

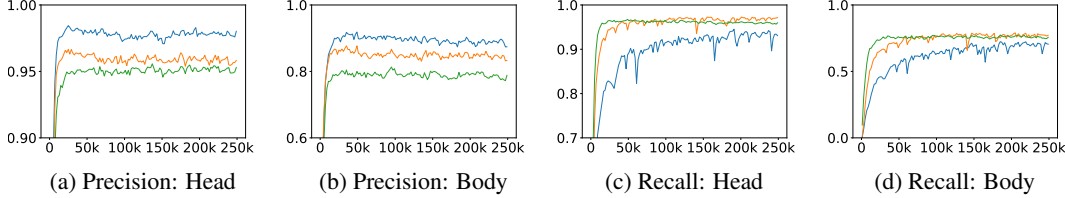

| (a) Precision: Head | (b) Precision: Body | (c) Recall: Head | (d) Recall: Body |

Figure B: **Precision-Recall Analysis on Head and Body Classes**: Orange and green curves denote FixMatch with threshold 0.95 and 0.6 respectively. InPL is denoted by blue curves. InPL consistently achieves higher pseudo-label precision with slightly lower recall compared with the confidence-based pseudo-labeling baselines.

|  | FixMatch (Sohn et al., 2020) | | | | InPL (ours) |
|---|---|---|---|---|---|
| Confidence threshold | $\tau = 0.95$ | $\tau = 0.8$ | $\tau = 0.7$ | $\tau = 0.6$ | - |
| Accuracy | 73.73 | 74.12 | 71.53 | 73.55 | **77.03** |

Table A: Comparison to FixMatch with various confidence thresholds on CIFAR10-LT. Results are generated with one run of 10% labeled data and imbalance ratio 100 with the same random seed.

## A.6 More Results on Robustness to Real OOD Samples

We present experimental results of InPL under the scenario of class mismatch in unlabeled data (Oliver et al., 2018). Specifically, we use six animal classes (bird, cat, deer, dog, frog, horse) of CIFAR10 in labeled data and only evaluate the model on these classes. The unlabeled data comes from four classes with different mismatch ratios. For example, with a 50% mismatch ratio, only two of the four classes in the unlabeled data are among the six classes in the labeled data.

As shown in Figure Ca, when the mismatch ratio is low, all methods with strong augmentation perform similarly. However, when the mismatch ratio becomes larger, InPL consistently achieves better performance. When unlabeled data contains OOD examples from different datasets (Table 5), InPL significantly outperforms FixMatch. Figure Cb also shows that InPL consistently pseudo-labels less OOD examples in training.

| Dataset | CIFAR10-LT | | | CIFAR100-LT |
|---|---|---|---|---|
| Imbalance Ratio | $\gamma = 100$ | $\gamma = 150$ | $\gamma = 200$ | $\gamma = 20$ |
| ReMixmatch + ABC-InPL w/o AML | $83.1_{\pm 0.88}$ | $80.8_{\pm 1.03}$ | $78.9_{\pm 1.14}$ | $57.5_{\pm 0.56}$ |
| ReMixmatch + ABC-InPL | $83.6_{\pm 0.45}$ | $81.3_{\pm 0.83}$ | $78.8_{\pm 0.75}$ | $58.4_{\pm 0.25}$ |

Table B: **Impact of AML on the ABC framework**.

## A.7 Contribution of Adaptive Margin Loss

Finally, we investigate the contribution of the adaptive margin loss (AML) over vanilla cross-entropy loss in our approach InPL. Within the FixMatch framework, the impact of AML can be derived from Table 1 where AML brings an additional 1-2% point improvement in overall accuracy. For the ABC framework, we use ReMixmatch + ABC-InPL as an example. As shown in Table B, AML generally brings in less than 1% point accuracy improvement on top of the energy-based pseudo-labeling.

## A.8 Results on Standard SSL Benchmarks

While designed to overcome the limitations of confidence-based pseudo-labeling on imbalanced data, InPL does not make any explicit assumptions about the class distribution. Thus, a natural question is: how does InPL perform on balanced SSL benchmarks? To answer this, we integrate InPL into

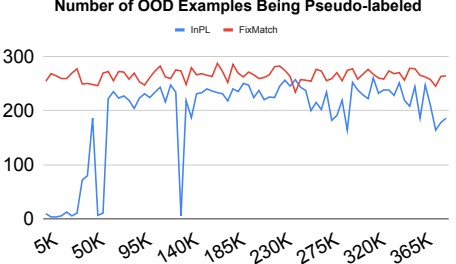

(a) Comparison of test error on CIFAR-10 (six animal classes) with different class mismatch ratio.

(b) Number of true OOD examples included

Figure C: **(a): Comparison of test error on CIFAR-10 (six animal classes) with different class mismatch ratio**. For example, "50%" indicates that two of the four classes in the unlabeled data are not present in the labeled data. InPL outperforms other state-of-the-art methods when the class mismatch ratio is high. **(b) Number of true OOD examples pseudo-labeled.** InPL consistently includes less true OOD examples compared with FixMatch and FlexMatch.

FixMatch (Sohn et al., 2020) and evaluate it on the balanced CIFAR10 (Krizhevsky & Hinton, 2009), SVHN (Netzer et al., 2011), and STL-10 (Coates et al., 2011) benchmarks using the default FixMatch training settings.

Table C summarizes the results. When the amount of labeled data is small (CIFAR $N = 40$ and SVHN $N = 40$), InPL shows a noticeable improvement over FixMatch, UDA, and DASH (Xu et al., 2021). This shows that the energy-based pseudo-labeling can still be useful in low-data regimes where confidence estimates may be less reliable even if the training data is balanced over classes. In other settings with more data (STL-10 $N = 1000$), InPL does not significantly outperform confidence-based methods such as FixMatch, but still shows competitive performance.

| | CIFAR10 | SVHN | STL-10 |
|---|---|---|---|
| | $N = 40$ | $N = 40$ | $N = 1000$ |
| UDA | $89.38_{\pm 3.75}$ | $94.88_{\pm 4.27}$ | $93.36_{\pm 0.17}$ |
| DASH† | $86.78_{\pm 3.75}$ | $96.97_{\pm 1.59}$ | $92.74_{\pm 0.40}$ |
| FixMatch | $92.53_{\pm 0.28}$ | $96.19_{\pm 1.18}$ | $93.75_{\pm 0.03}$ |
| FixMatch-InPL (ours) | $\mathbf{94.58}_{\pm 0.42}$ | $\mathbf{97.78}_{\pm 0.05}$ | $\mathbf{93.82}_{\pm 0.11}$ |

Table C: Top-1 accuracy on CIFAR10, SVHN, and STL-10. †Due to adaptation difficulties, we report the results of DASH from its original paper (Xu et al., 2021), which uses a different codebase. All methods (including ours) use the same backbone across experiments.

## A.9 THEORETICAL COMPARISON BETWEEN CONFIDENCE SCORE AND ENERGY SCORE

We borrow the analysis from prior work (Liu et al., 2020) to show the theoretical comparison between confidence scores and energy scores. The energy score for an in-distribution data point gets pushed down when the model is trained with the negative log-likelihood (NLL) loss (LeCun et al., 2006). This can be justified by the following derivation. For an in-distribution data $x$ with label $y$, with the energy definition from Equation 3, the gradient can be expressed as:

$$\frac{\partial \mathcal{L}_{\text{nll}}(x, y; \theta)}{\partial \theta} = \frac{1}{T} \frac{\partial E(x, y)}{\partial \theta} - \frac{1}{T} \sum_{j=1}^{K} \frac{\partial E(x, y)}{\partial \theta} \frac{e^{-E(x,y)/T}}{\sum_{j=1}^{K} e^{-E(x,j)/T}}$$
$$= \frac{1}{T} \Big( \underbrace{\frac{\partial E(x, y)}{\partial \theta}(1 - p(Y = y|x))}_{\downarrow \text{ energy pushed down for } y} - \underbrace{\sum_{j \neq y} \frac{\partial E(x, j)}{\partial \theta} p(Y = j|x)}_{\uparrow \text{ energy pulled up for other labels}} \Big).$$

The energy for label $y$ gets pushed down whereas the energy for other labels are pulled up. Since the energy is dominated by the label $y$, during training, the overall energy is pushed down by the NLL loss for in-distribution data.

The softmax score has connections to the energy score. The log of max softmax confidence can be decoupled as the energy score and the maximum logit value:

$$\log \max_y p(y \mid x) = E(x; f(x) - f^{\max}(x))$$

$$= \underbrace{E(x; f)}_{\downarrow \text{ for in-dist } x} + \underbrace{f^{\max}(x)}_{\uparrow \text{ for in-dist } x},$$

The energy score gets pushed down during training whereas the maximum logit term gets pulled up. These two conflicting trends make the softmax score unreliable to distinguish in-distribution and out-of-distribution examples.

In this work, we propose to treat the pseudo-labeling process as an evolving in-distribution and out-of-distribution classification problem where in-distribution examples are defined jointly with human-labeled data and pseudo-labeled data. Following this direction, we choose the energy-score as our pseudo-labeling criterion.

