# OpenReview forum: "InPL: Pseudo-labeling the Inliers First for Imbalanced Semi-supervised Learning"
_ICLR.cc/2023/Conference — ICLR 2023 poster_

### Official Review · Reviewer_EW6x · 2022-10-24

**Confidence:** 4
**Correctness:** 3
**Technical Novelty And Significance:** 3
**Empirical Novelty And Significance:** Not applicable
**Recommendation:** 6

**Clarity, Quality, Novelty And Reproducibility:**

* The paper is clearly written.

* While simple, it shows significant empirical improvement.

* The paper contains sufficient detail for reproducibility.

**Strength And Weaknesses:**

* Strength
  - The paper is written clearly.
  - The proposed method is very simple and shown to be effective. While simple, I tend to believe it is more than incremental, as it demonstrates consistent improvement across multiple SSL settings.

* Weakness
  - The comparison between the energy-based scoring and the confidence-based scoring should be more in-depth. Especially, the negative energy score $T \log \sum_{i=1}^{K} \exp(\frac{f_{i}(x)}{T})$ approximates a max logit $\max_{i=1}^{K} f_{i}(x)$ when there is a dominant one. Given such similarity, it is somewhat surprising that this simple change leads to such a huge difference.
    - The energy formulation is not invariant w.r.t the addition of values to logits, i.e., $E(f(x)) \neq E(f(x) + \mathrm{constant})$, while confidence is, i.e., $\mathrm{confidence}(f(x)) = \mathrm{confidence}(f(x) + \mathrm{constant})$. Would this difference be one of the reasons why energy formulation is better than the confidence, as the magnitude of the embedding is taken into account?
    - One interesting control experiment is to use a max-logit-based pseudo labeling, as it is similar to energy score in the sense that it is variant to the addition to logits.

**Summary Of The Paper:**

The paper introduces a new pseudo-labeling method based on the energy instead of a confidence for its application to imbalanced semi-supervised learning. Experimental results show improvement over existing methods.

**Summary Of The Review:**

The paper presents a simple trick that works well across SSL benchmarks. The paper could be improved with more in-depth analysis on why it works better, but overall I think it is a simple method that would benefit the community.

---

> ### Author Response · Authors · 2022-11-16
> **Author's Response**
>
> We thank you for your time and effort in reviewing our paper and providing thoughtful feedback to improve our paper. We are encouraged to see that you find our method simple and effective and also appreciate our clear writing.
>
>
> **C: In depth analysis between the energy-based scoring vs confidence-based scoring**
>
> The reason why the energy score is more reliable than the confidence score is because it provides a cleaner separation between in-distribution and out-of-distribution data. In our case, this translates to selecting more reliable unlabeled samples to pseudo-label.  We elaborate with more detailed theoretical and empirical justifications below.
>
> From a theoretical point of view, during training, the negative log-likelihood (NLL) loss pushes down the model’s energy for in-distribution data [1, 2]. To see this, borrowing the analysis from [2], the gradient of the loss with respect to the model’s parameters can be expressed as follows:
>
> \begin{align*}
>     \frac{\partial \mathcal{L}_\text{nll}(\*x,y; \theta)}{\partial \theta} &= \frac{1}{T}\frac{\partial E(\*x,y)}{\partial \theta} -  \frac{1}{T} \sum^K_\text{j=1} \frac{\partial E(\*x,y)}{\partial \theta} \frac{e^{-E(\*x,y)/T}}{\sum_\text{j=1}^K e^{-E(\*x,j)/T}} \\\ &= \frac{1}{T}\big(\underbrace{\frac{\partial E(\*x,y)}{\partial \theta}(1-p(Y=y) | \*x)}_\text{\downarrow~{energy push down for y}} - \underbrace{\sum_\text{j $\neq$ y} \frac{\partial E(\*x,j)}{\partial \theta} p(Y=j | \*x)}_\text{\uparrow {energy pull up for other labels}}
> \end{align*}
>
> The energy for label y gets increased whereas the energy for other labels gets decreased. During training, the energy is dominated by label y, therefore, the overall energy gets decreased.
>
> The softmax confidence score has connections to the energy score. Specifically, when T=1, the log of confidence score (max softmax score) can be decoupled as the energy score and the maximum logit value:
>
> \begin{align*}
> \log \max_y p(y \mid \*x) &= \underbrace{E(\*x;f)}_\text{\downarrow{ for in-dist $\*x$}}  + \underbrace{f^{{max}}(\*x)}_\text{\uparrow{ for in-dist $\*x$}}
> \end{align*}
>
> The energy score gets decreased during training whereas the maximum logit term gets increased. **These two conflicting trends make the softmax score unreliable for distinguishing in-distribution and out-of-distribution examples**.
>
> In this work, we propose to view pseudo-labeling as an evolving in-distribution vs out-of-distribution classification problem. The in-distribution is jointly formed by human-labeled and pseudo-labeled data at any training iteration and the unlabeled data that does not contribute to training can be thought as “out-of-distribution examples”. With this perspective, the confidence score is an unreliable indicator and we propose to use the energy score as the pseudo-label indicator.
>
> **From an empirical point of view**, fixed confidence thresholding results in low-precision of head pseudo-labels: even with a high confidence threshold (e.g, 0.95), the model still wrongly produces head pseudo-labels for tail classes (Section 4.4 and Appendix A.4). This demonstrates the unreliability of confidence score from empirical aspects.
>
> On the other hand, our approach improves pseudo-label precision for head classes alleviating the model bias towards head classes. Meanwhile, the recall for tail pseudo-labels is significantly improved without hurting the precision much.
>
> This simplicity makes our approach easy to integrate into the state-of-the-art SSL frameworks and achieve significant improvement (+3-4 % accuracy over vanilla FixMatch and +1-2 % accuracy over vanilla ABC on CIFAR10-LT).  This strength is recognized by other reviewers.
>
>
> [1] Lecun et al. A Tutorial on Energy-based Learning. MIT Press 2006
>
> [2] Liu et al, Energy-based Out-of-distribution Detection. Neurips 2020

---

> > ### Author Response · Authors · 2022-11-16
> > **Response Continued**
> >
> > **C: What about max-logit-based pseudo-labeling**
> >
> > Thank you for this suggestion! We conducted experiments of using max-logit as the pseudo-label criteria with different thresholds. The results are shown below on CIFAR10-LT with 10% labeled data and imbalance ratio 100 with the same random seed.
> >
> > |                | Max-Logit (threshold = 1) | Max-Logit (threshold = 2) | Max-Logit (threshold = 5) | Max-Logit (threshold = 10) | (Max-Logit (threshold = 12) | Confidence Score(threshold = 0.95) | Energy Score(threshold = -9.5) |
> > |----------------|---------------------------|---------------------------|---------------------------|---------------------------|----------------------------|------------------------------------|--------------------------------|
> > | Top-1 Accuracy | 73.11                     | 72.56                     | 73.66                     | 76.63                     | 73.26                      | 73.73                              | **77.03**                          |
> >
> > As shown in the table, with a proper threshold on max-logits, the model can achieve improvement over vanilla confidence-based pseudo-labeling yet does not show significant advantages over energy-based pseudo-labeling in our paper.

---

> ### Author Response · Authors · 2022-12-03
> **Looking forward to Discussion!**
>
> Dear Reviewer EW6x,
>
> Hope you are doing well and thanks for your effort again. We would like to highlight our response and check with you if all your questions have been properly answered.
>
> In our response, we:
>
> * provided theoretical analysis between the energy score and the confidence score
>
> * Experimented with your suggestion of using max logit as the criteria.
>
> Please do not hesitate to let us know if there is anything we can provide and thanks again for your review effort!

---

### Official Review · Reviewer_x1wG · 2022-10-25

**Confidence:** 3
**Correctness:** 4
**Technical Novelty And Significance:** 2
**Empirical Novelty And Significance:** 3
**Recommendation:** 3

**Clarity, Quality, Novelty And Reproducibility:**

The clarity, quality, and reproducibility of this work is good, but the novelty and contribution are limited to some extent.

**Strength And Weaknesses:**

Strengths

1.	The idea is simple and clear, and the paper is easy to understand and follow.

2.	This paper studies an important problem in imbalanced semi-supervised learning, i.e., the reliability of the pseudo labels.

Weaknesses

1.	The authors claim that the confidence-based methods are not reliable in imbalance semi-supervised learning, but it is not clear why the proposed method is reliable. The proposed method and the existing methods both depend on pre-defined thresholds.

2.	The motivation of the method is not very strong, and the contribution of this work is limited. The authors only change the criterion to pseudo-labeling an instance with energy score without proposing a new SSL approach.


**Summary Of The Paper:**

This paper studies imbalanced semi-supervised learning. SOTA imbalanced semi-supervised learning methods are often based on pseudo-labeling and consistency regularization, which still relies on confidence thresholding. In this paper, the authors formulate the pseudo-labeling problem as a classification problem, i.e., using the energy score to determine where an instance is in distribution or out of distribution. The authors claim that the proposed approach is more reliable compare to existing ones. Empirical results demonstrate the effectiveness of the proposed method.

**Summary Of The Review:**

Generally speaking, this paper proposes a simple method for imbalance semi-supervised learning, but the proposed method is not new. The authors did not give a sufficient discussion about the proposed method.

---

> ### Author Response · Authors · 2022-11-16
> **Author's Response**
>
> Thank you for your effort in reviewing our paper and providing constructive feedback. We are glad to see that you find the problem we study important, our idea clear and simple, and our paper easy to follow. We provide detailed responses to your comments below.
>
> **C: Contribution and novelty**
>
> We agree that the energy score in itself is not something new that we propose in this paper. However, it is more important to understand why the energy score, a metric that has never been used in pseudo-labeling previously, can serve as a good indicator of the quality of pseudo-labels in SSL. In this sense, we believe our method is new.
>
> To the best of our knowledge, this is the first work exploring a non-confidence-based pseudo-labeling process. Importantly, we are the first to study the unreliability of pseudo-labels in the imbalanced scenario (as you had mentioned under ``strengths’’).
>
> In terms of empirical performance, our method significantly improves SSL performance under the imbalanced scenario (Tables 1, 2, 3 and 4), expresses some level of robustness when OOD examples are present in the unlabeled data (Tables 5, Figures C in Appendix A.6), and still achieves reasonable performance on standard SSL benchmarks (Table 4 and Table C in Appendix).
>
> **C: The motivation is not very strong**
>
> We are happy to restate the motivation of our work.
>
> **Problem Statement**. The drawback of using a fixed high confidence threshold in imbalanced SSL comes in two aspects:
> It results in extremely low-recall for tail class pseudo-labels
> It results in low precision for head class pseudo-labels because the model still wrongly predicts tail instances as head pseudo-labels with an extremely high confidence (passing the threshold)
>
> We updated our introduction (Section 1) to make the problem statement more clear and the corresponding analysis can be found in Section 4.4.
>
> **Limitation of confidence threshold**. We show that using a lower confidence threshold can solve problem 1 but it significantly hurts the overall precision (Section 4.4). Problem 2 comes from the natural property of the confidence score: neural networks can predict high confidence even on out-of-distribution examples.  In imbalanced SSL, the model only sees few samples from tail classes, and hence tends to incorrectly pseudo-label true unlabeled tail class samples as head classes. Doing so results in reduced precision for pseudo-labels in head classes, and in turn lowers precision of pseudo-labels over all classes (see Figures 4 and B).
>
> **Our solution**. Given the analysis above, we argue that confidence score may not be the optimal pseudo-label criterion for imbalanced SSL and propose to view the pseudo-labeling process as an in-distribution vs out-of-distribution classification problem. This way, we only pseudo-label unlabeled examples that are close to the in-distribution (and hence whose pseudo labels are more likely to be reliable) and the in-distribution gradually expands and evolves during training. To serve this goal, we adopt the energy score as it has been proven as an effective metric for out-of-distribution (OOD) detection.
>
> We hope our explanation clarifies the motivation of our work. Feel free to let us know if there’s still any confusion.

---

> > ### Author Response · Authors · 2022-11-16
> > **Response Continued**
> >
> > **C: Why the proposed method is reliable when both the proposed threshold and confidence-based threshold use fixed threshold.**
> >
> > The reason why the energy score is more reliable than the confidence score is because it provides a cleaner separation between in-distribution and out-of-distribution data. In our case, this translates to selecting more reliable unlabeled samples to pseudo-label.  We elaborate with more detailed theoretical and empirical justifications below.
> >
> > From a theoretical point of view, during training, the negative log-likelihood (NLL) loss pushes down the model’s energy for in-distribution data [1, 2]. To see this, borrowing the analysis from [2], the gradient of the loss with respect to the model’s parameters can be expressed as follows:
> >
> > \begin{align*}
> >     \frac{\partial \mathcal{L}_\text{nll}(\*x,y; \theta)}{\partial \theta} &= \frac{1}{T}\frac{\partial E(\*x,y)}{\partial \theta} -  \frac{1}{T} \sum^K_\text{j=1} \frac{\partial E(\*x,y)}{\partial \theta} \frac{e^{-E(\*x,y)/T}}{\sum_\text{j=1}^K e^{-E(\*x,j)/T}} \\\ &= \frac{1}{T}\big(\underbrace{\frac{\partial E(\*x,y)}{\partial \theta}(1-p(Y=y) | \*x)}_\text{\downarrow~{energy push down for y}} - \underbrace{\sum_\text{j $\neq$ y} \frac{\partial E(\*x,j)}{\partial \theta} p(Y=j | \*x)}_\text{\uparrow {energy pull up for other labels}}
> > \end{align*}
> >
> > The energy for label y gets increased whereas the energy for other labels gets decreased. During training, the energy is dominated by label y, therefore, the overall energy gets decreased.
> >
> > The softmax confidence score has connections to the energy score. Specifically, when T=1, the log of confidence score (max softmax score) can be decoupled as the energy score and the maximum logit value:
> >
> > \begin{align*}
> > \log \max_y p(y \mid \*x) &= \underbrace{E(\*x;f)}_\text{\downarrow{ for in-dist $\*x$}}  + \underbrace{f^{{max}}(\*x)}_\text{\uparrow{ for in-dist $\*x$}}
> > \end{align*}
> >
> > The energy score gets decreased during training whereas the maximum logit term gets increased. **These two conflicting trends make the softmax score unreliable for distinguishing in-distribution and out-of-distribution examples**.
> >
> > In this work, we propose to view pseudo-labeling as an evolving in-distribution vs out-of-distribution classification problem. The in-distribution is jointly formed by human-labeled and pseudo-labeled data at any training iteration and the unlabeled data that does not contribute to training can be thought as “out-of-distribution examples”. With this perspective, the confidence score is an unreliable indicator and we propose to use the energy score as the pseudo-label indicator.
> >
> > **From an empirical point of view**, fixed confidence thresholding results in low-precision of head pseudo-labels: even with a high confidence threshold (e.g, 0.95), the model still wrongly produces head pseudo-labels for tail classes (Section 4.4 and Appendix A.4). This demonstrates the unreliability of confidence score from empirical aspects.
> >
> > On the other hand, our approach improves pseudo-label precision for head classes alleviating the model bias towards head classes. Meanwhile, the recall for tail pseudo-labels is significantly improved without hurting the precision much.
> >
> > This simplicity makes our approach easy to integrate into the state-of-the-art SSL frameworks and achieve significant improvement (+3-4 % accuracy over vanilla FixMatch and +1-2 % accuracy over vanilla ABC on CIFAR10-LT).  This strength is recognized by other reviewers.
> >
> >
> > [1] Lecun et al. A Tutorial on Energy-based Learning. MIT Press 2006
> >
> > [2] Liu et al, Energy-based Out-of-distribution Detection. Neurips 2020

---

> ### Author Response · Authors · 2022-12-03
> **Looking forward to Discussion!**
>
> Dear Reviewer x1wG,
>
> Hope you are doing well and thanks for your effort again. We would like to highlight our response and check with you if all your questions have been properly answered.
>
> In our response, we:
>
> * Clarified our contribution and novelty.
>
> * Restated our motivation (we also updated our manuscript)
>
> * Provided theoretical analysis on why energy score is a better criteria
>
> Please do not hesitate to let us know if there is anything we can provide and thanks again for your review effort!

---

### Official Review · Reviewer_b7AL · 2022-10-25

**Confidence:** 4
**Correctness:** 4
**Technical Novelty And Significance:** 4
**Empirical Novelty And Significance:** 3
**Recommendation:** 6

**Clarity, Quality, Novelty And Reproducibility:**

Despite minor concerns, the proposed method is clear and novel. The quality of this paper looks favorable overall.

**Strength And Weaknesses:**

Strength
- The authors propose a new perspective to tackle the imbalanced semi-supervised learning problem, and the proposed solution is original.
- The proposed method shows favorable performance compared to the recent methods.

Weakness
- It would be better to improve the explanation of the motivation. For example, Fig. 3 says that the energy-based criterion selects fewer incorrect samples for the pseudo-labels. However, it is hard to see that in the figure. It would be better to provide a quantitative measure to show precisely how many incorrect samples are selected with which criterion.
- Also, instead of discarding the fixmatch-based criterion, how about using both criteria? In other words, use the pseudo-labels when the confidence is above a certain threshold, and the energy is below a certain threshold.
- More importantly, there is no analysis of the design choice for the proposed method. Instead of simply borrowing the popular energy function, it would be better to analyze how to modify the loss function or what is the best hyper-parameter (at least about the energy threshold) for the best performance.
- Finally, there is no evaluation result on realistic datasets such as iNaturalist or ImageNet datasets. Otherwise, it is hard to verify that the proposed method is helpful for realistic scenarios.


**Summary Of The Paper:**

The authors present a new perspective of pseudo-labeling for imbalanced SSL.
Without relying on model confidence, they propose to measure whether an unlabeled sample is likely to be “in-distribution,”; i.e., close to the current training data.
To decide whether an unlabeled sample is “in-distribution” or “out-of-distribution,” the authors adopt the energy score from out-of-distribution detection literature.
As training progresses and more unlabeled samples become in-distribution and contribute to training, the combined labeled and pseudo-labeled data can better approximate the accurate class distribution to improve the model.


**Summary Of The Review:**

Despite several concerns about the experiments, the observation in this work is interesting, and the proposed method is novel.

---

> ### Author Response · Authors · 2022-11-16
> **Authors' Response**
>
> Thank you for your effort in reviewing our paper and providing thoughtful comments. We are encouraged that you find our approach to be original and recognize the improvement of our method. Please see our responses below:
>
> **C: Improve the explanation of motivation**
>
> We are happy to restate the motivation of our work.
>
> **Problem Statement**. The drawback of using a fixed high confidence threshold in imbalanced SSL comes in two aspects:
> It results in extremely low-recall for tail class pseudo-labels
> It results in low precision for head class pseudo-labels because the model still wrongly predicts tail instances as head pseudo-labels with an extremely high confidence (passing the threshold)
>
> We updated our introduction (Section 1) to make the problem statement more clear and the corresponding analysis can be found in Section 4.4.
>
> **Limitation of confidence threshold**. We show that using a lower confidence threshold can solve problem 1 but it significantly hurts the overall precision (Section 4.4). Problem 2 comes from the natural property of the confidence score: neural networks can predict high confidence even on out-of-distribution examples.  In imbalanced SSL, the model only sees few samples from tail classes, and hence tends to incorrectly pseudo-label true unlabeled tail class samples as head classes. Doing so results in reduced precision for pseudo-labels in head classes, and in turn lowers precision of pseudo-labels over all classes (see Figures 4 and B).
>
> **Our solution**. Given the analysis above, we argue that confidence score may not be the optimal pseudo-label criterion for imbalanced SSL and propose to view the pseudo-labeling process as an in-distribution vs out-of-distribution classification problem. This way, we only pseudo-label unlabeled examples that are close to the in-distribution (and hence whose pseudo labels are more likely to be reliable) and the in-distribution gradually expands and evolves during training. To serve this goal, we adopt the energy score as it has been proven as an effective metric for out-of-distribution (OOD) detection.
>
> We hope our explanation clarifies the motivation of our work. Feel free to let us know if there’s still any confusion.
>
> **C: Quantitative results are not shown in Figure 3**
>
> Thank you for this suggestion. Figure 3 serves as an illustrative comparison between confidence-based thresholding and energy-based thresholding. A detailed precision-recall comparison result can be found in Figure 4 and Appendix A.4. The results show that our method achieves better overall precision and better precision for head pseudo-labels with significantly improved recall for tail pseudo-labels.
>
>
> **C: How about using both criteria**
>
> Interesting suggestion!  We have experimented with this idea before. Specifically, we combine the best energy threshold with different confidence thresholds on CIFAR10-LT with 10% labels and imbalance ratio 100:
>
> |                | Energy + Confidence (confidence threshold= 0.6) | Energy + Confidence (confidence threshold = 0.7) | Energy + Confidence (confidence threshold = 0.8) | Energy + Confidence (confidence threshold = 0.9) | Energy + Confidence (confidence threshold = 0.95) | Confidence only (confidence threshold = 0.95) | Energy only (energy threshold = -9.5) |
> |----------------|-------------------------------------------------|--------------------------------------------------|--------------------------------------------------|--------------------------------------------------|---------------------------------------------------|-----------------------------------------------|---------------------------------------|
> | Top-1 Accuracy | 76.10                                           | 76.43                                            | 76.13                                            | 76.54                                            | 75.95                                             | 73.73                                         | **77.03**                                 |
>
> As shown in the table, using both energy and confidence score as pseudo-label criteria does not improve the performance of using only the energy score. The energy score itself is already a strong pseudo-label criteria for imbalanced SSL.

---

> > ### Author Response · Authors · 2022-11-16
> > **Response Continued**
> >
> > **C: Analysis of design choice for the proposed method**
> >
> > Thank you for the comments.
> >
> > The analysis on threshold choice, which we set via cross-validation, can be found in Appendix A.3 and the ablation on the marginal loss can be found in Appendix A.8. Under the FixMatch framework, choosing the energy threshold to be between -8 and -10 generally results in strong performance whereas the best result is obtained at -9.5. For the ABC framework, our method is not very sensitive to the energy threshold. Most threshold values result in good performance but a slightly higher threshold of -5.5 achieves the best accuracy. The AML loss generally brings in less than 1% accuracy improvement and our method achieves significant improvement over vanilla confidence-based counterparts without it (Table 1).
> >
> > Additionally, we provide precision and recall analysis for head/body classes in Appendix A.4, where our method significantly improves the pseudo-label precision. Our method also achieves competitive performance when true OOD examples exist in the unlabeled data and our method consistently includes less OOD examples in training without making explicit assumptions (Sections 4.5 and A.6).
> >
> >
> > **C: Evaluation on realistic datasets**
> >
> > We agree that evaluating on large-scaled datasets is important. We present results of our method on ImageNet-127 following imbalanced SSL and ImageNet following standard SSL settings. We report the unweighted average of recall across classes since the test set is imbalanced following prior work.
> >
> > Due to limited computation resources, we cannot match the large batch sizes used by FixMatch and CReST (1024 for labeled data and 5120 for unlabeled data). Instead, we use a batch size of 128 for both labeled and unlabeled data, which took around 9 days per run. For fair comparison, we also re-run FixMatch with this setting and the results are shown in the table below.  These results are also included in our revised manuscript in Section 4.3 and Table 4.
> >
> >
> > |                      | ImageNet-127    | ImageNet  |
> > |----------------------|-------|-------|
> > | FixMatch             | 51.96 | 56.34 |
> > | FixMatch-InPL (Ours) | **54.82** | **57.92** |
> >
> >
> > As shown in the table, compared with confidence-based pseudo-labeling in FixMatch, our method achieves better performance on both ImageNet-127 (imbalanced) and ImageNet (balanced).

---

> ### Author Response · Authors · 2022-12-03
> **Looking forward to Discussion!**
>
> Dear Reviewer b7AL,
>
> Hope you are doing well and thanks for your effort again. We would like to highlight our response and check with you if all your questions have been properly answered.
>
> In our response, we:
>
> * Restated the motivation of our work (we also updated our manuscript for this part)
>
> * Explained the purpose of Figure 3. In short, Figure 3 only serves for an illustrative comparison and quantitative results can be found in Figure 4 and B
>
> * Experimented with your suggestion of using both criteria.
>
> * Added results on large-scale datasets (also added to our manuscript)
>
> Please do not hesitate to let us know if there is anything we can provide and thanks again for your review effort!

---

### Official Review · Reviewer_YGBu · 2022-10-27

**Confidence:** 4
**Correctness:** 4
**Technical Novelty And Significance:** 2
**Empirical Novelty And Significance:** 3
**Recommendation:** 6

**Clarity, Quality, Novelty And Reproducibility:**

**Clarity and Quality**. Overall, the writing is very clear and easy to follow. In addition, the organization of the main draft is well-established.

**Novelty**. While the proposed idea is a new thing in the imbalanced SSL, it has been investigated in the other relevant literature; hence, this work is limited in technical novelty.

**Reproducibility**. The authors provide a detailed presentation of the experimental setups, hence it seems to be easy to reproduce the results.



**Strength And Weaknesses:**

**Pros.**

- **Simple and effective solution for an important problem**. Considering the class imbalance scenario is an essential step for applying SSL in a more realistic scenario but has yet to be explored. The proposed method can be used with a simple modification of the existing baseline, and it shows significant empirical gain; hence, it has the potential to be widely used to mitigate this problem without additional burdens.

**Cons.**

- **Omitted baselines in robust semi-supervised literature**. The proposed idea, which considers the imbalanced SSL under the OOD detection framework, is interesting and somewhat new. However, at the same time, there are concerns about its technical novelty since many relevant works in robust SSL literature have yet to be mentioned and compared [1,2,3,4]. Overall, these works detect the OOD samples in unlabeled data using their score functions; while there are no works that use energy score, but using the energy score to detect the OOD samples is not a new one [5]. Hence, in some sense, the proposed method could be viewed as a combination of existing techniques for the new problem. So, it seems to be necessary to compare with the previous robust SSL methods as well.
- **Absence of large-scale experiments.** Previous imbalanced SSL works have proposed the experimental results on large datasets such as ImageNet-127 [6] or LSUN [7]; however, INPL has only been demonstrated under relatively small benchmarks (CIFAR-LT-10 and 100). For a real-world application, experiments on such large benchmarks are essential.

[1] Nair et al., RealMix: Towards Realistic Semi-Supervised Deep Learning Algorithms., arXiv:19.12

[2] Guo et al., Safe Deep Semi-Supervised Learning for Unseen-Class Unlabeled Data., ICML 20

[3] Saito et al., OpenMatch: Open-set Consistency Regularization for Semi-supervised Learning with Outliers., NeurIPS 21

[4] Park et al., OpenCoS: Contrastive Semi-supervised Learning for Handling Open-set Unlabeled Data., ECCV 22 (w)

[5] Liu et al., Energy-based Out-of-distribution Detection., NeurIPS 20

[6] Wei et al., CReST: A Class-Rebalancing Self-Training Framework for Imbalanced Semi-Supervised Learning., CVPR 21

[7] Lee et al., ABC: Auxiliary Balanced Classifier for Class-imbalanced Semi-supervised Learning., NeurIPS 21

**Summary Of The Paper:**

This paper tackles an imbalanced semi-supervised learning (SSL) problem by proposing a new way to construct the pseudo labels of unlabeled samples, coined INlier Pseudo-Labeling (INPL). Unlike the previous approaches, which use the softmax confidence from the training classifier, INPL instead uses the energy score calculated by logsumexp of logits. As INPL only changes the score function for unlabeled samples, it is compatible with the existing imbalance or semi-supervised learning methods. Also, it does not incur additional computational costs. Significant empirical improvements on widely used imbalanced SSL benchmarks (CIFAR-10-LT and CIFAR-100-LT) and more general scenarios (existence of OOD samples in unlabeled data or balanced class distribution) demonstrate the effectiveness of the proposed method.

**Summary Of The Review:**

As aforementioned, this paper is quite limited in the aspect of technical novelty. However, the empirical results are significant, and the presentation is clear. Hence, I believe that the acceptance of this work will be a useful addition to the ICLR community.

---

> ### Author Response · Authors · 2022-11-16
> **Authors' Response**
>
> Thank you for your effort in reviewing our paper. We are encouraged that you find the problem we study meaningful and our proposed method simple and effective.
>
> We also appreciate your constructive feedback to improve our paper. We incorporated them into the revised version of our manuscript, which can be found in the system.
> We provide detailed responses to your comments below:
>
> **C: Omitted baselines in Robust SSL literature.**
>
> Thank you for this suggestion. We agree that existing work in robust SSL literature should be discussed and compared.
>
> Following your suggestion, we integrated OpenMatch [1] and D3SL [2] into our codebase with their official open-source implementation and evaluated on the CIFAR10-LT benchmark. For fair comparison, we add the weak-strong augmentation paradigm to D3L. We report the mean over 3 different runs. These results are also included in our revised manuscript in Section 4.1 and Table 1.
>
>
> |                      | $\gamma$ = 50    | $\gamma$ = 100  | $\gamma$ = 200   |
> |----------------------|-------|-------|-------|
> | FixMatch             | 80.84 | 72.95 | 63.25 |
> | OpenMatch            | 74.27 | 62.33 | 57.35 |
> | OpenMatch(Detach)    | 81.01 | 73.15 | 63.22 |
> | D3SL                 | 81.20 | 72.71 | 65.09 |
> | FixMatch-InPL (Ours) | **83.36** | **76.05** | **66.47** |
>
> InPL outperforms the robust SSL methods, OpenMatch and D3L. When training on the closed-set imbalance scenarios, we find that the outlier detection module in OpenMatch causes convergence issues and results in suboptimal performance. When applying stop gradient to the outlier detection module (“OpenMatch (detach)”), the performance of OpenMatch can match FixMatch yet still falls behind our method InPL.
>
> We are also running experiments on CIFAR100-LT but these experiments take longer time to finish. We added the results of CIFAR10-LT to Table 1 and will update it with CIFAR100-LT results once they are ready.
>
> [1] Salto et al, OpenMatch: Open-set Consistency Regularization for Semi-supervised Learning with Outliers. Neurips 2021
>
> [2] Guo et al, Safe Deep Semi-Supervised Learning for Unseen-Class Unlabeled Data. ICML 2020
>
>
> **C: Absence of large scale experiments**
>
> We agree that evaluating on large-scaled datasets is important. We present results of our method on ImageNet-127 following imbalanced SSL and ImageNet following standard SSL settings. We report the unweighted average of recall across classes since the test set is imbalanced following prior work.
>
> Due to limited computation resources, we cannot match the large batch sizes used by FixMatch and CReST (1024 for labeled data and 5120 for unlabeled data). Instead, we use a batch size of 128 for both labeled and unlabeled data, which took around 9 days per run. For fair comparison, we also re-run FixMatch with this setting and the results are shown in the table below.  These results are also included in our revised manuscript in Section 4.3 and Table 4.
>
>
> |                      | ImageNet-127    | ImageNet  |
> |----------------------|-------|-------|
> | FixMatch             | 51.96 | 56.34 |
> | FixMatch-InPL (Ours) | **54.82** | **57.92** |
>
>
> As shown in the table, compared with confidence-based pseudo-labeling in FixMatch, our method achieves better performance on both ImageNet-127 (imbalanced) and ImageNet (balanced).
>
> **C: Novelty of this work**
>
> Thank you for bringing this up! We are more than happy to have this discussion.
>
> We agree that the energy score in itself is not something new that we propose in the paper. However, it is more important to understand why the energy score, a metric that has never been used in pseudo-labeling previously, can serve as a good indicator of the quality of pseudo-labels in SSL which improves the pseudo-label recall for tail classes and the overall pseudo-label precision (Section 4.4). Although we did not propose this metric, we provide a new perspective of the pseudo-labeling process (“in-dist vs out-of-dist classification”), which has not been explored before as discussed in Section 1.
>
> Please feel free to follow up with any further questions. We are happy to provide our responses.

---

### Author Response · Authors · 2022-11-16
**General Response and Updated Manuscript**

We thank all reviewers for their review efforts and constructive feedback to improve our paper. We are glad that Reviewers **YGBu** and **EW6x** find our method to be simple and effective and that the novelty of our work is recognized by Reviewer **b7AL**. We are also encouraged that Reviewer **x2WG** acknowledges the importance of the problem we study in this paper.


Our revision of the manuscript, marked in red in the pdf, is summarized as follows:

* Results on large-scale datasets (Reviewer **YGBu** and **b7AL**). We evaluate our method on the large-scale ImageNet-127 and ImageNet datasets and our method shows favorable improvements over confidence-based counterparts.

* Discussing and comparing with robust SSL baselines. We run additional experiments and demonstrate that these robust SSL methods do not show strong performance under class imbalance in comparison to our method. (Reviewer **YGBu**).

* Restating the motivation of our work in the introduction (Reviewer **b7AL** and **x1wG**).

We hope that our responses adequately address all suggestions, and welcome any further feedback during the discussion.

Paper 1095 Authors

---

### Decision · Program_Chairs · 2023-01-20

**Decision:**

Accept: poster

**Justification For Why Not Higher Score:**

The simple approach, IPL, works well for several SSL datasets. Simple, neat, and solid approach that works is always a good addition to the conference. Nonetheless, presenting it as a poster is a proper decision, given that the energy-based criterion is leveraged from the well-established outlier detection field to the SSL field.

**Justification For Why Not Lower Score:**

Reviewers are in general consensus of acceptance. While one reviewer has a lower score (3: Reject), the concerns were addressed reasonably well after the AC's investigation.

**Metareview: Summary, Strengths And Weaknesses:**

This paper proposes an interesting approach, Inlier Pseudo-labeling (IPL), towards the challenging imbalanced semi-supervised learning (SSL) problem. A nontrivial insight is observed by the authors: To decide whether an unlabeled sample is "in-distribution'' or "out-of-distribution'' can be beneficial to imbalanced SSL. The proposed approach is based on the energy score from out-of-distribution detection literature (LeCun, 2006). While simple, the IPL approach achieves significantly better performance compared to the standard method, FixMatch. When combined with other more recent SoTA methods, the improvement is much smaller but still significant. A strength is that the IPL works well on the large-scale ImageNet dataset.

Paper received rather controversial comments at first. Authors actively prepared the rebuttal, providing with most new results and elaborations requested by the reviewers. After discussion, three reviewers are on the positive side, while only one reviewer is on the negative side. AC carefully read the paper, the reviews, the revised version, and the rebuttal, and acknowledge that the response has addressed the open concerns reasonably well. Considering that introducing the energy-based criterion to SSL is new, and the simple IPL works generally well for imbalanced SSL datasets, AC believes the paper is worth to be seen by the SSL community at the conference.

In addition to the current revised version, authors shall integrate all rebuttal material into the paper, especially the elaboration on the motivation of the energy-based score and why it is beneficial.

**Note From Pc:**

if the above contains the word "oral" or "spotlight" please see: "oral" presentation means -> notable-top-5% and "spotlight" means -> notable-top-25%. As stated in our emails, we are disassociating presentation type from AC recommendations